# Temperature Variation at Solid-Fluid Interface of Thin Film Lubricated Contact Problems

**Szabolcs Szávai [1],\* and Sándor Kovács [2]**

[1]   Institute of Machine and Product Design, Faculty of Mechanical Engineering and Informatics,
      University of Miskolc, 3515 Miskolc, Hungary
[2]   Bay Zoltán Nonprofit Ltd. for Applied Research, 3519 Miskolc, Hungary; sandor.kovacs@bayzoltan.hu
**\***   Correspondence: szavai.szabolcs@uni-miskolc.hu

**Abstract:** Many calculating methods have been already developed for solving contact problems of parts such as gears, cams, and followers under fluid film lubrication conditions considering the temperature and pressure dependence. Similarly, the determination of the elasto-hydrodynamic pressure distribution the processes taking place in the lubricant and the contacting bodies, as well as in their environment, have to be dealt with simultaneously for the determination of the temperature field. A system of equation for the modelling of thermo-elastohydrodynamic lubrication between two contacting bodies containing hydrodynamic, thermodynamic, and strength problems is a highly non-linear system which becomes even more so if the temperature and pressure dependence of the material properties are considered. To solve this system, scientists started to use the finite element formulation in the 1960s and it was found to be a promising and reliable method. Earlier, the lubrication analysts used only the h-version finite element method (h-FEM) till 1991, when the first usage of the p-version finite element method (p-FEM) was published in the literature. In order to reduce the problem, in case of point or line contact, the contact bodies can be handled as semi-infinite ones. Following this simplification that had been successfully applied for the gap size determination, a substructure model was defined using analytical solution of the moving heat source. Instead of an iterative way between the solid and fluid problem, in this paper we present an efficient solution when thermal model for lubricant and surfaces were coupled and solved by a direct numerical method in one step.

**Keywords:** TEHD; lubrication; FEM; temperature; variation; thin; film; simulation

---

## 1. Introduction

For rolling sliding parts such as cams and followers, gears, and bearings often operate under high loads, high speed, and high slip; the local or global temperature rise caused by the heat dissipation generated by the pressure distribution acting on the surfaces and the tangential stresses developing in the lubricant, respectively, may reach a level resulting in a non-negligible deformation of the surfaces as well as influencing the lubricant properties. Sternlicht et al. [1] (1961) investigated thermal effects in elasto-hydrodynamic lubricated (EHL) contacts. Cheng and Sternlicht [2] (1965) assumed that the viscosity is constant through the oil film and solved the thermal EHL model. Later, Cheng [3] (1965) obtained a refined solution in which the TEHL model got variable viscosity through the film thickness. Murch and Wilson [4] (1975) proved the significant effect of thermal conditions on minimum film thickness applying high rolling speeds.

Sadeghi [5] (1990) published a complete numerical solution for TEHD (thermal elasto-hydrodynamic) problem using different slip effects on the contact area with the same conclusions as Murch and Wilson.

Similarly to the determination of the elasto-hydrodynamic pressure distribution, the processes taking place in the lubricant and the contacting bodies, as well as in their environment, have to be dealt with simultaneously for the determination of the temperature field. Based on the research mentioned above, in our work only the inertia forces and turbulences are neglected from the general Reynolds equation. The thermodynamic model of the fluid–film contact has got non-constant temperature condition inside the film.

The scientists used finite difference method to obtain the solution of the generalized Reynolds equation. In case of these solving systems, a very high number of points of grid is needed. The finite element formulation is researched by Reddi [6] (1969), Reddi and Chu [7] (1970), Rohde [8] (1974), Rohde and Oh [9] (1975), Garcia-Suarez et al. [10] (1984), and Freund and Tieu (1993) [11].

These lubrication analysts used only the h-version finite element method (h-FEM) for solving Reynolds and generalized Reynolds equation till 1991, when the first usage of the p-version finite element method (p-FEM) was published in the literature (Nguyen [12]). In this work, the geometry and the lubricant properties were assumed to be constant and even today the p-FEM method is typically used in solid mechanics and heat transfer problems.

As is generally the case of the finite element method, the weak convergence of the approximation is searched which means that the Reynolds and energy equation was transformed to weighted-residual integral form. For the approximation Legendre polynomials were used to obtain the unknown pressure and temperature distribution. In our work, a simulation procedure was developed based on a finite element method to model thermo-hydrodynamic processes. This method is based on our EHD (elasto-hydrodynamic) simulation procedure, which was supplemented by the incorporation of discretized equations describing the thermal conditions. Using p-FEM (p-version of finite element method) for solving TEHD (thermo elasto-hydrodynamic) problems allows the replacement of the fine mesh with a coarse one. In case of p-FEM, a discretization strategy is used in which the finite element mesh is fixed and the degrees of approximating polynomials are increased.

As for additional development, it is shown that boundary condition can be redefined as all inlet and outlet can be adiabatic boundaries, therefore, they can be treated uniformly and there is no need to distinguish between them.

In order to reduce the problem, in case of point or line contact, the contact bodies can be handled as semi-infinite ones. In the interest of possessing the compatible form to finite element tensor expression, the gap size has been calculated as a sum of the sizes of the original geometry and deformation of a half-space in discretized form to which the displacement of a rigid surface has been added. The least square method can be used to obtain the coefficients of discretized form in order to approximate the analytical solution of deformation for half-space. Following this simplification that was successfully applied for the gap size determination, a substructure model was defined using an analytical solution of the moving heat source where the input temperature data were obtained from p-FEM calculation (Carslaw and Jaeger (1959) [13]).

Instead of an iterative way between the solid and fluid problem, in this paper we present an efficient solution when a thermal model for lubricant and surfaces were coupled and solved by a direct numerical method in one step.

Due to the coarse mesh which is usually used in the case of p-FEM, cavitation has to be managed inside elements in order to obtain an accurate solution of the cavitation. Typically in the literature, deactivating those elements where the cavitation appears is enough for the authors; however, only a lower level of accuracy of calculation can be obtained. For the above reasons, a new penalty parameter method was developed in which a density–pressure function is described based on work by Kumar and Booker. The error of approximation can be decreased using higher penalty values and the method is adaptable if cavitation pressure $p_c \neq 0$, as well. Unlike previous penalty parameter methods, continuity can be ensured here, so this method can be easily incorporated into the p-FEM model of TEHD.

## 2. Theoretical Description of the Elastohydrodynamic Lubrication

A general case of contacting surface pairs in the status of TEHL (thermal elastohydrodynamic lubrication) can be seen in Figure 1. The gap between the mating bodies is filled with lubricant because of to the relative velocity of the bodies. The developed hydrodynamic pressure is affected by the motion of the lubricant. The relative velocity difference between the mating surfaces results in tangential (shear) stress in the lubricant which cause the flow of the fluid material. Thus, at the proper kinematic condition of the contacting bodies and the pressure distribution in the lubricant can maintain balance with the clamping force on the contacting bodies and with this prevent a body-to-body contact. Pressure distribution acting on the surfaces may reach a level resulting in a non-negligible deformation of the surfaces. The pressure and the tangential stresses developing in the lubricant generate heat dissipation which cause local temperature rise influencing the lubricant properties. If the circumstances of contact developing underneath thermal elasto-hydrodynamic conditions of lubrication are needed to be modelled, then hydrodynamic, thermodynamic, and solid-structural problems must be solved simultaneously with a modelling system which is highly non-linear, even by itself, but also because of material properties. However, these three main areas may be separated clearly from each other in respect of their basic equations.

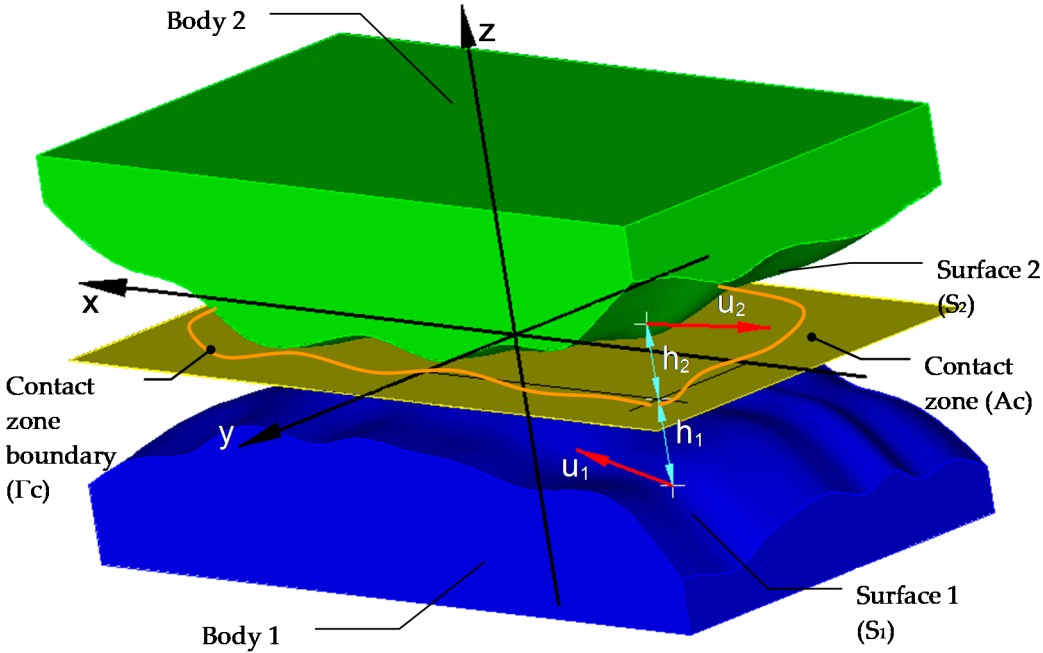

**Figure 1.** Contacting bodies.

For the contact problem of lubrication theory—due to its nature—it is convenient to employ in general a Gauss coordinate system with axis *z* perpendicular to the center contact surface.

Because of the nature of the contact problem of lubrication theory, it is generally advisable to use a Gauss coordinate system of which the *z* axis is perpendicular to the ideal center of the contact area. However, in the case of point or spot contact the center contact surface may be considered as a plane with its normal being parallel with the line of action of the force pressing the surfaces to each other, i.e., of the contact pressure. Consequently, it is most convenient to use for the investigation and description of the phenomenon an orthogonal coordinate system with its axis *z* being coincident with the line of action of the contact pressure developed by the compressive force.

### 2.1. Contact Pressure and Film Shape

The generalized Reynolds equation of Dowson (1961) [14] is highly non-linear partial differential equation with variable viscosity and density extents through the film thickness which is used to calculate the contact pressure generated by fluid film lubrication.

The gap height can be calculated as a sum of the sizes of the original geometry and deformation of a half-space to which the displacement of a rigid surface has been added [15]:

$$h = h_1 + h_2 = h_{g_1} + \Delta_{rigid1} + \delta_{1_1} + h_{g2} + \Delta_{rigid2} + \delta_2 = h_g + \Delta_{rigid} + \delta \tag{1}$$

where $\Delta_{rigid}$ is the displacement of rigid surface, $h_g$ is the original gap height, the $\delta_i$ deformation of the contacting solid bodies.

The exact details of the generalized Reynolds and displacement field equations and calculation with them can be found in literature [15,16] in order to determine the pressure and flow field of the liquid lubricant.

### 2.2. Penalty Cavitation

During the description of lubrication problems both the boundary conditions defined hypothetically and the limitations applicable to the free edge forming the boundary of the actual lubrication zone which developed due to the cavitation, respectively, were encountered simultaneously.

In the cavitation zone the lubricant flows adhered to the surfaces and were broken up into strips. For this reason, the magnitude of the gas filling factor must be taken into account also for the thermodynamical equation.

At the hypothetically assumed edge the primary equation definable for the pressure field:

$$p = p_a \ x, y \in \Gamma_c^p \tag{2}$$

can be identified on the one hand and the secondary boundary conditions defining the flow rate of lubricant exiting at the edge, on the other hand:

$$\vec{q}_h \cdot \vec{n}_{\Gamma_c} = q_h^{n_{\Gamma_c}} = q_a \ x, y \in \Gamma_c^q \tag{3}$$

where $p_a$ is the external pressure and $q_a$ the flow rate of lubricant exiting at the edge.

However, the lubrication region with the hypothetically assumed $\Gamma_c$ edge does not coincide in a significant portion of the cases with the region filled with unbroken lubricant film to 100%. The location of edge ($\Gamma_c^{cav}$) of the actual lubrication region is not known beforehand but can be determined on the basis of the properties applicable to it.

In case the lubricant film breaks down due to the effect of cavitation and the sub-cavitation pressure is neglected then the cavitational boundary condition introduced by Swift-Stieber [17], commonly called the Swift-Stieber boundary condition in the literature, should be satisfied at the primary and secondary boundary conditions inside the cavitation region and at its boundary:

$$p = p_c \left| \nabla_{xy} p \right| = 0 \ x, y \in \Gamma_c^q \tag{4}$$

where $p_c$ is the cavitational or saturation pressure.

While the satisfaction of the primary and secondary boundary conditions (2)–(3) does not entail any special difficulty, numerous problems are raised by the cavitational boundary condition (4). The difficulty in the resolution process is constituted primarily by the fact that the boundaries of the lubricating film broken down by cavitation are unknown. Consequently, the boundaries of the initially assumed region do not coincide with the real boundaries ($\Gamma_c$—Figure 2) of the lubricating film. Thus, the relationships written previously for a homogeneous phase continuous lubricating film cannot be applied in the entirety of the presumed region bounded by $\Gamma_c$.

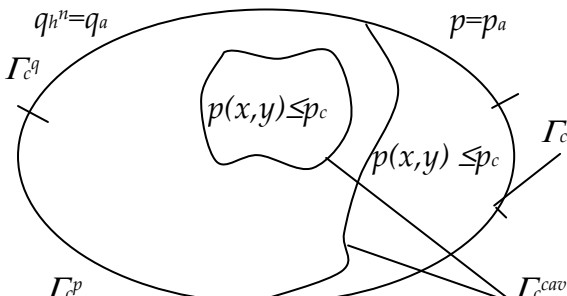

**Figure 2.** Boundaries of the contact zone and the cavitation edge.

The Elrod-Adams algorithm is widely used for extending the Reynolds equation inside the cavitation zone which includes the $\theta = \rho/\rho_c$ fractional film content [18]. Kumar and Booker [19,20] published the method which is fitted for the FEM procedure. This method separates the density and pressure determination. Using of linear correlation between the density and the viscosity was also suggested by Kumar and Booker [19,20]:

$$\frac{\eta}{\eta_L} = \frac{\rho}{\rho_L} \quad \rho \leq \rho_c \tag{5}$$

Instead of the separation of the variables for the contact and cavitation zone, penalty cavitation method has been proposed by Szávai [16] where the pressure dependent density in the cavitation zone is approximated by a high gradient slope in case of sufficiently low pressure (i.e., under saturation pressure). With these conditions the density can be described as follows [16]:

$$\rho^{*:} = \frac{\rho_L(p, \vartheta)}{\gamma(p) \cdot (p_c - p) + 1} \tag{6}$$

where $\gamma(p)$ is the penalty function which is $\gamma(p) = c$ if $p < p_c$ otherwise *0*, where *c* is a sufficiently high number.

The pressure dependency of the density exists both in the lubrication region and cavitation zone. Thus, $\rho^*$ range contains the cavitation zone and the lubrication region as well. The volume fraction based on (6) can be obtained as [16]:

$$\theta(p) = \frac{\rho^*}{\rho_L} = \frac{1}{\gamma(p)(p_c - p) + 1} \tag{7}$$

Applying assumption of Kumar and Booker of the density and viscosity correlation the viscosity can be obtained as [16]:

$$\eta^{*:} = \eta_L(p, \vartheta) \frac{\rho^*}{\rho_L(p, \vartheta)} = \theta(p) \eta_L(p, \vartheta) \tag{8}$$

### 2.3. Kinematic Properties of Lubricant in the Gap

The $F(\tau_{eq})$ is a characteristic function of a specific type of a lubricant model and $\tau_{eq}$ is the equivalent shear stress in $\tau_{eq} = \sqrt{\frac{1}{2}\sigma' \cdots \sigma'}$ where $\sigma'$ is the stress deviator tensor.

In the case of various types of lubricant models, the function $F(\tau_{eq})$ can be equal to the forms below [17]:

$$F(\tau_{eq})_{Newton} = \frac{\tau_{eq}}{\eta}; F(\tau_{eq})_{Eyring} = \frac{\tau_E}{\eta}\sinh\left(\frac{\tau_{eq}}{\tau_E}\right); F(\tau_{eq})_{viscoplastic} = -\frac{\tau_L}{\eta}\ln\left(1 - \frac{\tau_{eq}}{\tau_L}\right);$$
$$F(\tau_{eq})_{simple\_viscoplastic} = -\frac{\tau_{eq}}{\eta}\left(1 - \left|\frac{\tau_{eq}}{\tau_L}\right|\right)^{-1}; F(\tau_{eq})_{circular} = \frac{\tau_{eq}}{\eta}\left[1 - \left(\frac{\tau_{eq}}{\tau_L}\right)^2\right]^{-\frac{1}{2}} \tag{9}$$

where $\tau_E$ is the Eyring shear stress and $\tau_L$ is the limit shear stress.

$\tau_L$ can be taking account as a linear function of pressure of which coefficients $\tau_{l0}$ and $\chi$:

$$\tau_L = \tau_{l_0} + \chi p \tag{10}$$

Since boundary conditions for the velocity field on the surfaces—as shown in Figure 1—are:

$$\begin{aligned}
z = -h_1 : \vec{u} = \vec{u}_1 = [u_1, v_1, w_1] = \left[\vec{u}_{xy_1}, w_1\right] \\
z = h_2 : \vec{u} = \vec{u}_2 = [u_2, v_2, w_2] = \left[\vec{u}_{xy_2}, w_2\right]
\end{aligned} \tag{11}$$

$$\left(\frac{\partial \vec{u}_{xy}}{\partial z}\right)^* = \frac{1}{\theta} \frac{F(\tau_e)}{\tau_e}\left[(\nabla_{xy} p)\left(z - \frac{F_1}{F_0}\right) + \theta\frac{\left(\vec{u}_{xy_1} - \vec{u}_{xy_1} - \vec{K}_{0xy}\right)}{F_0}\right] + A\frac{d\vec{\sigma}'_z}{dt} \tag{12}$$

$$\begin{aligned}
\left(\vec{u}_{xy}\right)^* = \vec{u}_{xy1} + \frac{1}{\theta}\nabla_{xy} p \left(\int_{-h_1}^{z} \frac{F(\tau_{eq})}{\tau_e} z\, dz - \frac{F_1}{F_0}\cdot\int_{-h_1}^{z} \frac{F(\tau_{eq})}{\tau_e}\, dz\right) + \\
+ \frac{\vec{u}_{xy2} - \vec{u}_{xy1} - \vec{K}_{0xy}}{F_0}\cdot\int_{-h_1}^{z}\frac{F(\tau_{qe})}{\tau_{eq}}\, dz + \int_{-h_1}^{z} A\frac{d\vec{\sigma}'_z}{dt}\, dz
\end{aligned} \tag{13}$$

$$w^* = w = w_1 + \frac{w_2 - w_1}{F_0}\int_{-h_1}^{z}\frac{F(\tau_{eq})}{\tau_e}\, dz \tag{14}$$

where $F_0 = \int_{-h_1}^{h_2}\frac{F(\tau_{eq})}{\tau_{eq}}\, dz$   $F_1 = \int_{-h_1}^{h_2}\frac{F(\tau_{eq})}{\tau_{eq}} z\, dz$   $\vec{K}_{0xy} = \int_{-h_1}^{h_2} A\frac{d\vec{\sigma}'_z}{dt}\, dz = \begin{bmatrix}\int_{-h_1}^{h_2}\frac{Ad\tau_{xz}}{dt}\, dz \\ \int_{-h_1}^{h_2}\frac{Ad\tau_{yz}}{dt}\, dz\end{bmatrix}$ according to [15], and

$\vec{\sigma}'_z = \sigma' \cdot \vec{e}_z = \begin{bmatrix}\tau_{xz} \\ \tau_{yz}\end{bmatrix}$ and $A$ is compressibility (=1/K where $K$ is bulk modulus).

The velocity component in $z$ direction can be decomposed as [15]:

$$\begin{aligned}
w_1 = -\vec{u}_{xy_1}\left(\nabla_{xy} h_1\right) + W_1(t, x, y) \\
w_2 = \vec{u}_{xy_2}\left(\nabla_{xy} h_2\right) + W_2(t, x, y)
\end{aligned} \tag{15}$$

where $W_i$ is rigid body motion of the contacting body in $z$ direction, while the first member comes from movement of non-plane surface in $x$ and $y$ direction.

The filling parameter taken into consideration the derivative of velocity along the gap and the stress in this direction may be written in the following form considering that the value of the filling parameter in the contact region is 1 and the pressure in the cavitation zone is negligibly low, respectively:

$$\left(\frac{\partial \vec{u}_{xy}}{\partial z}\right)^* \approx \frac{F(\tau_e)}{\tau_e}\left[(\nabla_{xy} p)\left(z - \frac{F_1}{F_0}\right) + \frac{\left(\vec{u}_{xy2} - \vec{u}_{xy1} - \vec{K}_{0xy}\right)}{F_0}\right] + A\frac{d\vec{\sigma}'_z}{dt} = \frac{\partial \vec{u}_{xy}}{\partial z} \tag{16}$$

$$\begin{aligned}
\left(\vec{u}_{xy}\right)^* \approx \vec{u}_{xy1} + \nabla_{xy} p\left(\int_{-h_1}^{z}\frac{F(\tau_{eq})}{\tau_e} z\, dz - \frac{F_1}{F_0}\cdot\int_{-h_1}^{z}\frac{F(\tau_{eq})}{\tau_e}\, dz\right) + \\
+ \frac{\vec{u}_{xy2} - \vec{u}_{xy1} - \vec{K}_{0xy}}{F_0}\cdot\int_{-h_1}^{z}\frac{F(\tau_{qe})}{\tau_{eq}}\, dz + \int_{-h_1}^{z} A\frac{d\vec{\sigma}'_z}{dt}\, dz = \vec{u}_{xy}
\end{aligned} \tag{17}$$

$$\vec{\sigma}'_z = \frac{\tau_{eq}}{F(\tau_{eq})}\left(\frac{\partial \vec{u}_{xy}}{\partial z} - A\frac{d\vec{\sigma}'_z}{dt}\right) \tag{18}$$

$$\vec{\sigma}'_z{}^* = \left( (\nabla_{xy} p)\left(z - \frac{F_1}{F_0}\right) + \theta \frac{\left(\vec{u}_{xy_2} - \vec{u}_{xy_1 1} - \vec{K}_{0xy}\right)}{F_0} \right) \approx$$
$$\approx \theta \left( (\nabla_{xy} p)\left(z - \frac{F_1}{F_0}\right) + \frac{\left(\vec{u}_{xy_2} - \vec{u}_{xy_1} - \vec{K}_{0xy}\right)}{F_0} \right) = \theta \vec{\sigma}'_z \tag{19}$$

The stress deviator tensor takes the simplified form below for Newtonian fluids:

$$\vec{\sigma}'_z = \eta \frac{\partial \vec{u}_{xy}}{\partial z} \tag{20}$$

## 3. Theoretical Description of Thermodynamical Part of the TEHD

The next equation expresses the conservation of energy stating that the change in total energy during unit time is equal to the sum [16]:

$$\frac{d(c_v \vartheta)}{dt} + p\left(\nabla \cdot \vec{u}\right) = \frac{1}{\rho}\nabla \cdot (\lambda \nabla \vartheta) + \frac{\Xi}{\rho} \tag{21}$$

where $\Xi$ is the dissipation originating in the internal friction of the lubricant:

$$\Xi = \boldsymbol{\sigma}' \cdot\cdot \mathbf{D} \tag{22}$$

$\vartheta$—the temperature, and
$c_v$—specific heat referred to constant volume.
**D**—strain rate
**σ′**—deviator stress tensor
If the thermal properties of the continuum are presumed to be constant, then:

$$c_v \frac{d\vartheta}{dt} + p\left(\nabla \cdot \vec{u}\right) = \frac{1}{\rho}\nabla \cdot (\lambda \nabla \vartheta) + \frac{\Xi}{\rho} \tag{23}$$

for an uncompressible liquid ($\nabla \cdot \vec{u} = 0$)

$$c_v \left( \frac{\partial \vartheta}{\partial t} + \vec{u} \cdot (\nabla \vartheta) \right) = \frac{1}{\rho}\nabla(\lambda \nabla \vartheta) + \frac{\Xi}{\rho} \tag{24}$$

Similarly to the determination of the elasto-hydrodynamic pressure distribution the processes taking place in the lubricant and the contacting bodies as well as in their environment have to be dealt with simultaneously for the determination of the temperature field. The details of the processes taking place in the contacting bodies will not be discussed here, but the assumption employed according to which the temperature fields developing in the bodies can be determined analytically or numerically on the basis of the boundary conditions applicable to the bodies, primarily in respect of temperature, and secondarily in respect of heat transfer.

Because of the nature of the problem the following simplification may be employed similarly to the simplifications seen at the derivation of the Reynolds equation according to [17]:

$$\nabla \cdot (\lambda \nabla \vartheta) \approx \frac{\partial}{\partial z}\left( \lambda \frac{\partial \vartheta}{\partial z} \right) \tag{25}$$

However, this does not mean a significant easement for the solution since the temperature variation along the $z$ axis cannot be eliminated. For the calculation of the temperature field the temperature variation along the gap also has to be determined unless the approach assuming parabolic temperature distribution along the cross-section is employed. In this latter case it is sufficient to determine a typical

temperature distribution (mean or average temperature) independent of the gap thickness coordinate (*z*) if the temperatures of the contacting bodies are known or determined by other equations.

Under the conditions of elasto-hydrodynamic lubrication—as also outlined before—the flow may be considered to be laminar as the gap size is very small, barely a few μm. Thus—with turbulence neglected—the terms averaged in time may be considered to constitute the parameters usable for describing the fluid motion.

The dissipation in the lubricant is:

$$\Xi = \tau_{xz} \cdot \left( \frac{\partial u}{\partial z} + \frac{\partial w}{\partial x} \right) + \tau_{yz} \cdot \left( \frac{\partial v}{\partial z} + \frac{\partial w}{\partial y} \right) \tag{26}$$

Taking into consideration that the derivatives along the gap of the *u* and *v* fields are significantly larger than the derivative of the w field along the contact plane the dissipation will be the following with negligible error [21]:

$$\Xi \approx \tau_{xz} \cdot \frac{\partial u}{\partial z} + \tau_{yz} \cdot \frac{\partial v}{\partial z} = \vec{\sigma}'_z \cdot \frac{\partial \vec{u}_{xy}}{\partial z} \tag{27}$$

Thus, the dissipation can be written in the following form for Newtonian fluids:

$$\Xi \approx \eta \left[ \left( \frac{\partial u}{\partial z} \right)^2 + \left( \frac{\partial v}{\partial z} \right)^2 \right] = \eta \left( \frac{\partial \vec{u}_{xy}}{\partial z} \right)^2 \tag{28}$$

### 3.1. Thermal Boundary Conditions

At variance with the boundary condition for pressure boundary conditions have to be defined for the temperature field both at the contact surfaces ($S_1$, $S_2$) and for the entire cross-section of the oil film at the boundary of the contact zone.

Both surfaces $S_1$ and $S_2$ can be divided into two sections ($S_{i\vartheta}$, $S_{iQ}$) enabling thereby to specify a primary boundary condition for the temperature field, on the one hand, and a secondary boundary condition defining the convection of heat through the surfaces:

$$\vartheta(x, y, h_i) = \vartheta_{s_i}(x, y) \ (x, y) \in S_{i\vartheta} \tag{29}$$

$$\frac{\partial \vartheta}{\partial n_{S_i}} = \nabla \vartheta \cdot \vec{n}_S = \frac{1}{\lambda} \vec{q}_{S_i}(x, y) \cdot \vec{n}_S = -\frac{1}{\lambda} q_{S_i}(x, y)(x, y) \in S_{iQ} \tag{30}$$

where $\vartheta_{si}(x,y)$ is the temperature of surface $S_{i\vartheta}$ while $q_{si}$ is the convection of heat through surface $S_{iQ}$ which are either pre-defined or determined during the solution of the associated thermal problem (relationship between the lubricant and the contacting body).

Oil inlet and outlet boundaries have free temperature condition ($q_n = 0$).

In the case of pure sliding the contact region moves together with one of the surfaces or—worded in another way—this surface is stationary in relation to the contact region. In this case the heat exchange taking place through the surface stationary in relation to the contact zone can be neglected in most cases since it is more intensive by several orders of magnitude through the surface of the body moving in relation to the contact zone. At this time the adiabatic boundary condition proposed by Rohde and Oh [8] can also be applied to the surface moving with contact region:

$$\frac{\partial \vartheta}{\partial n_s} = \nabla \vartheta \cdot \vec{n}_s \approx \frac{\partial \vartheta}{\partial z} = 0 \tag{31}$$

The adiabatic modelling of this surface makes problem solving significantly easier, for example in the case of the quasi-stationary line contact of bodies regarded as infinite semi-spaces as the temperature distribution of the body moving with the contact region cannot be determined analytically or only in a very complex manner beyond the initial transient phase.

While generally accepted boundary conditions (30) and (31) have been developed for contacting surfaces $S_1$ and $S_2$ during the evolution of this field of science different boundary conditions may be encountered at the inlet and outlet cross-section of the lubricating film. An applicable approximation could involve the consideration of the derivatives of the temperature of the lubricant films carried on the surfaces taken in parallel with the plane of the surface to be negligibly small and the free surface of the lubricant in contact with air to be adiabatic with good approximation as it was presented in [16]

$$\frac{\partial \vartheta(x,y,z)}{\partial n_{\Gamma_c}} = 0; \ r(x,y) \in \Gamma_c, \ -h_1 \le z \le h_2 \tag{32}$$

A boundary condition similar to this was seen also in previous calculations [17] as well since no other reliable and practically applicable boundary condition could be defined because of the problems associated with backflow and the determination of the thickness of the lubricant film carried on the surfaces and of the initial position of the contact region in connection with this.

### 3.2. Consideration of Cavitation

In the cavitation zone the lubricant flows adhering to the surfaces and broken up into strips. For this reason, the magnitude of the gas filling factor must be taken into account also for the thermodynamical equation. Thus, similarly to (6) and (8)—assuming that condition (4) is satisfied in the hydrodynamic problem—the dissipation according to (26)–(28) will be with a negligible error [16]:

$$\Xi^* \approx \vec{\sigma}'_z \left(\frac{\partial \vec{u}_{xy}}{\partial z}\right)^* \approx \theta \vec{\sigma}'_z \frac{\partial \vec{u}_{xy}}{\partial z} = \theta \Xi \tag{33}$$

The coefficient of thermal conductivity varies as the function of density like the viscosity, thus [16]:

$$\lambda^{*:} = \theta \lambda_L \tag{34}$$

but, at the same time, the coefficient of specific heat does not change because of its specific nature:

$$c_v^{*:} = c_{vL} = c_v. \tag{35}$$

On the basis of the above, the differential equation of thermal conductivity (23)–(24) takes the modified form shown below:

$$\frac{d(c_v \vartheta)}{dt} + p \nabla \cdot \vec{u} = \frac{1}{\theta \rho_L} \nabla(\theta \lambda_L \cdot \nabla \vartheta) + \frac{\Xi_L}{\rho_L}. \tag{36}$$

If the thermal parameters of the continuum are considered to be constant:

$$c_v \frac{d\vartheta}{dt} + p \nabla \cdot \vec{u} = \frac{\lambda_L}{\theta \rho_L} \nabla(\theta \cdot \nabla \vartheta) + \frac{\Xi_L}{\rho_L}. \tag{37}$$

Since, according to the law of the conversion of mass:

$$\frac{d\rho^*}{dt} + \rho^* \nabla \cdot \vec{u} = 0. \tag{38}$$

Taking (7) into consideration term *div(u)* in Equation (37) of thermal conductivity may be written in the following form:

$$\nabla \cdot \vec{u} = -\frac{1}{\rho^{\bullet}} \frac{d\rho^*}{dt} = -\frac{\frac{\partial \rho^*}{\partial t} + \nabla \rho^* \cdot \vec{u}}{\rho^*} = -\frac{\frac{\partial \rho_L}{\partial t} + \nabla \rho_L \cdot \vec{u}}{\rho_L} - \frac{\frac{\partial}{\partial t}\theta + \nabla \theta \cdot \vec{u}}{\theta} \tag{39}$$

On this basis:

$$\frac{d(c_v\vartheta)}{dt} + p\left(-\frac{\frac{\partial\rho_L}{\partial t} + \nabla\rho_L\cdot\vec{u}}{\rho_L} - \frac{\frac{\partial}{\partial t}\theta + \nabla\theta\cdot\vec{u}}{\theta}\right) = \frac{1}{\theta\rho_L}\nabla(\theta\lambda_L\cdot\nabla\vartheta) + \frac{\Xi_L}{\rho_L} \tag{40}$$

or in another form, expanding the substantial derivative:

$$\frac{\partial(c_v\vartheta)}{\partial t} + \nabla(c_v\vartheta)\cdot\vec{u} + p\left(-\frac{\frac{\partial\rho_L}{\partial t} + \nabla\rho_L\cdot\vec{u}}{\rho_L} - \frac{\frac{\partial}{\partial t}\theta + \nabla\theta\cdot\vec{u}}{\theta}\right) = \frac{1}{\theta\rho_L}\nabla(\theta\lambda_L\cdot\nabla\vartheta) + \frac{\Xi_L}{\rho_L} \tag{41}$$

Let us multiply the above equation with $\theta\rho_L$ in order to obtain symmetric matrixes in the finite element model:

$$\theta\rho_L\left(\frac{\partial(c_v\vartheta)}{\partial t} + \nabla(c_v\vartheta)\cdot\vec{u}\right) + p\left(-\frac{\frac{\partial\rho_L}{\partial t} + \nabla\rho_L\cdot\vec{u}}{\rho_L} - \frac{\frac{\partial}{\partial t}\theta + \nabla\theta\cdot\vec{u}}{\theta}\right)\theta\rho_L =$$
$$= \nabla(\theta\lambda_L\cdot\nabla\vartheta) + \theta\Xi_L \tag{42}$$

The above equation may be simplified as follows in some basic cases.

If the lubricant density is constant ($\rho_L = const$, $d\rho_L/dt = 0$), then:

$$-\frac{\frac{\partial\rho_L}{\partial t} + \nabla\rho_L\cdot\vec{u}}{\rho_L} - \frac{\frac{\partial}{\partial t}\theta + \nabla\theta\cdot\vec{u}}{\theta} = -\frac{\frac{\partial}{\partial t}\theta + \nabla\theta\cdot\vec{u}}{\theta}. \tag{43}$$

In the case of stationary contact:

$$-\frac{\frac{\partial\rho_L}{\partial t} + \nabla\rho_L\cdot\vec{u}}{\rho_L} - \frac{\frac{\partial}{\partial t}\theta + \nabla\theta\cdot\vec{u}}{\theta} = -\frac{\nabla\rho_L\cdot\vec{u}}{\rho_L} - \frac{\nabla\theta\cdot\vec{u}}{\theta}. \tag{44}$$

At stationary contact with constant lubricant density:

$$-\frac{\frac{\partial\rho_L}{\partial t} + \nabla\rho_L\cdot\vec{u}}{\rho_L} - \frac{\frac{\partial}{\partial t}\theta + \nabla\theta\cdot\vec{u}}{\theta} = -\frac{\nabla\theta\cdot\vec{u}}{\theta}. \tag{45}$$

Returning to original Equation (42) $\rho_L = \rho_c$ may be considered to be approximately constant in the cavitation zone. Above the lubrication region, however, the value of the filling parameter is $\theta = 1$. Thus, in the lubrication zone:

$$-\frac{\frac{\partial\rho_L}{\partial t} + \nabla\rho_L\cdot\vec{u}}{\rho_L} - \frac{\frac{\partial}{\partial t}\theta + \nabla\theta\cdot\vec{u}}{\theta} = -\frac{\frac{\partial\rho_L}{\partial t} + \nabla\rho_L\cdot\vec{u}}{\rho_L} \tag{46}$$

while in the cavitation zone

$$-\frac{\frac{\partial\rho_L}{\partial t} + \nabla\rho_L\cdot\vec{u}}{\rho_L} - \frac{\frac{\partial}{\partial t}\theta + \nabla\theta\cdot\vec{u}}{\theta} = -\frac{\frac{\partial}{\partial t}\theta + \nabla\theta\cdot\vec{u}}{\theta}. \tag{47}$$

Let us notice that $\rho_L$ may vary only in the lubrication zone where $\theta = 1$ but, at the same time, may vary only in the cavitation zone where, however, $\rho_L = \rho_c$ with good approximation.

On this basis, the new thermal conductivity equation derived with the use of the filling parameter will be [16]:

$$\theta\rho_L\left(\frac{\partial(c_v\vartheta)}{\partial t} + \nabla(c_v\vartheta)\cdot\vec{u}\right) + p\left(-\frac{\partial\rho_L}{\partial t} - \nabla\rho_L\cdot\vec{u} - \left(\frac{\partial}{\partial t}\theta + \nabla\theta\cdot\vec{u}\right)\rho_c\right) =$$
$$= \nabla(\theta\lambda_L\cdot\nabla\vartheta) + \theta\Xi_L \tag{48}$$

The following simplified differential equation is obtained with constant density and thermal parameters:

$$\theta \rho_L c_v \left( \frac{\partial \vartheta}{\partial t} + \nabla \vartheta \cdot \vec{u} \right) - p \left( \frac{\partial}{\partial t} \theta + \nabla \theta \cdot \vec{u} \right) \rho_L = \lambda_L \nabla (\theta \cdot \nabla \vartheta) + \theta \Xi_L. \tag{49}$$

### 3.3. Temperature Variation of Contacting Bodies as the Result of Heat Sources Distributed on Their Surfaces

As presented in Section 3.1, determining the temperatures of the contacting bodies and their surfaces is a high priority task during the solution of the thermo-dynamical problem. As obvious also from boundary condition (29) the temperature of the lubricant adhering to the surfaces in the contact region is identical with the surface temperature and, furthermore, heat defined by (30) affects the bodies through the surfaces:

$$\begin{array}{ll} q_{S_1}(x,y) = \theta \lambda_L \cdot \frac{-\partial \vartheta(x,y,-h_1)}{\partial z}; & \vec{n}_{S_1} \approx -\vec{e} \\ q_{S_2}(x,y) = \theta \lambda_L \cdot \frac{\partial \vartheta(x,y,h_2)}{\partial z}; & \vec{n}_{S_2} \approx \vec{e}_z \end{array} \; ; \; (x,y) \in A_c. \tag{50}$$

Similarly to the determination of surface deformation, modelling the contacting bodies as semi-infinitive bodies is applicable frequently also for the calculation of the thermal part of thermo–elasto–hydrodynamic problems. If this can be done, then the relationships according to [13] elaborated for the case of heat sources moving in and infinite half-space (at a point in infinite space [13]) in the determination of the temperature distribution of the bodies:

$$\vartheta_s(x,y,z) - \vartheta_{S_i}^0 = \frac{1}{2\pi} \int\limits_{A(\hat{x},\hat{y})} \frac{q_{S_i}(\hat{x},\hat{y})}{\rho_{S_i} c_{S_i} \kappa_{S_i}} \frac{e^{\frac{\vec{r} \cdot \vec{u}_{xy,S_i} - U_{S_i}R}{2\cdot\kappa}}}{R} dA(\hat{x},\hat{y}) \tag{51}$$

$$\begin{array}{l} R^2 = (\hat{x} - x)^2 + (\hat{y} - y)^2 + z^2 \\ r^2 = (\hat{x} - x)^2 + (\hat{y} - y)^2 \\ U_{xy,S_i}^2 = \vec{u}_{xy,S_i} \cdot \vec{u}_{xy,S_i} \end{array} \tag{52}$$

while for line loads [16]:

$$\vartheta_{S_i}(x,z) - \vartheta_{S_i}^0 = \frac{1}{\pi} \int\limits_{s(\hat{x})} \frac{q_{S_i}(\hat{x})}{\rho_{S_i} c_{S_i} \kappa_{S_i}} e^{\frac{U_{S_i}(\hat{x}-x)}{2\cdot\kappa}} K_0 \left[ \frac{U_{S_i} \sqrt{(\hat{x}-x)^2 + z^2}}{2\kappa} \right] ds(\hat{x}) \tag{53}$$

where $K_0$ is a second type, zero order modified Bessel function.

Let us introduce the symbol shown below:

$$\Lambda_{S_i} = \frac{1}{2\pi} \frac{\lambda_{LS_i}}{\rho_{S_i} c_{S_i} \kappa_{S_i}} \frac{e^{\frac{\vec{r} \cdot \vec{u}_{S_i} - U_{S_i}r}{2\cdot\kappa_{S_i}}}}{r} \cdot \left( \vec{n}_{S_i} \cdot \vec{e}_z \right) \tag{54}$$

$$\vartheta_s(x,y,z) - \vartheta_{S_i}^0 = \int\limits_{A(\hat{x},\hat{y})} \frac{\partial \vartheta}{\partial z} \bigg|_{S_i} \theta \Lambda_{S_i} dA(\hat{x},\hat{y}) \tag{55}$$

which will take the following form in the same sense for line contact with (53) taken into consideration [16]:

$$\Lambda_{S_i} = \frac{1}{\pi} \frac{\lambda_{LS_i}}{\rho_{S_i} c_{S_i} \kappa_{S_i}} e^{\frac{U_{S_i} \cdot (\hat{x}-x)}{2\cdot\kappa_{S_i}}} K_0 \left[ \frac{U_{S_i}(\hat{x}-x)}{2\kappa_{S_i}} \right] \cdot \left( \vec{n}_{S_i} \cdot \vec{e}_z \right) \tag{56}$$

$$\vartheta_{S_i}(x,z) - \vartheta_{S_i}^0 = \int\limits_{s(\hat{x})} \frac{\partial \vartheta}{\partial z}\bigg|_{S_i} \theta \Lambda_{S_i} ds(\hat{x}) \tag{57}$$

If the above approximation cannot be permitted, then the substructure approach well-known in the finite element practice may be employed in which case the thermal problem of contacting bodies is handled as a substructure of tribology problem. This is less efficient but the combined determination of the temperature field applicable to both the structure and the contact region can also be carried out if the construction of the cause-and-effect matrix necessary for the sub-structural model cannot be solved or only difficultly.

### 3.4. Weak Integral form of the Thermodynamic Equation

The weak integral form of thermodynamic Equation (42) can be set up with the use of a weight function $w_Q$ [16]:

$$\int\limits_{V_c} w_Q \Bigg( \theta \rho_L \Big( \frac{\partial(c_v \vartheta)}{\partial t} + \nabla(c_v \vartheta) \cdot \vec{u} \Big) - p\Big( \frac{\partial \rho_L}{\partial t} + \rho_c \frac{\partial}{\partial t}\theta + \nabla \rho_L \cdot \vec{u} + \nabla \theta \cdot \vec{u} \rho_c \Big) - \theta \Xi_L \Bigg)$$
$$+ \theta \lambda \cdot \nabla w_Q \cdot \nabla \vartheta \Bigg) dV - \oint\limits_{\Omega c} \theta \lambda_L w_Q \nabla \vartheta \cdot \vec{n}_\Omega dA = 0 \tag{58}$$

on the basis of (30) equivalent to:

$$\int\limits_{V_c} w_Q \Bigg( \theta \rho_L \Big( \frac{\partial(c_v \vartheta)}{\partial t} + \nabla(c_v \vartheta) \cdot \vec{u} \Big) - p\Big( \frac{\partial \rho_L}{\partial t} + \rho_c \frac{\partial}{\partial t}\theta + \nabla \rho_L \cdot \vec{u} + \nabla \theta \cdot \vec{u} \rho_c \Big) - \theta \Xi_L \Bigg)$$
$$+ \theta \lambda_L \cdot \nabla w_Q \cdot \nabla \vartheta \Bigg) dV - \oint\limits_{\Omega c} w_Q \vec{q}_Q \cdot \vec{n}_\Omega dA = 0 \tag{59}$$

if vector $\boldsymbol{u}$ is introduced as a finite element symbol from $w_Q = \mathbf{N}_\vartheta$ and vector $u$, then:

$$\int\limits_{V_c} \theta \rho_L \frac{\partial c_v}{\partial t} \mathbf{N}_\vartheta \mathbf{N}_\vartheta^T dV \mathbf{T} + \int\limits_{V_c} \theta \rho_L c_v \mathbf{N}_\vartheta \mathbf{N}_\vartheta^T dV \frac{\partial \mathbf{T}}{\partial t} + \int\limits_{V_c} \theta \rho_L c_v \mathbf{N}_\vartheta \big( \mathbf{u}^T (\nabla \mathbf{N}_\vartheta^T) \big) dV \mathbf{T} +$$
$$\int\limits_{V_c} \theta \rho_L \mathbf{N}_\vartheta \big( \mathbf{u}^T (\nabla c_v) \mathbf{N}_\vartheta^T \big) dV \mathbf{T} - \int\limits_{V_c} p\Big( \frac{\partial \rho_L}{\partial t} + \rho_c \frac{\partial}{\partial t}\theta + \mathbf{u}^T (\nabla \rho_L + \rho_c \nabla \theta) \Big) \mathbf{N}_\vartheta dV -$$
$$- \int\limits_{V_c} \theta \Xi_L \mathbf{N}_\vartheta dV + \int\limits_{V_c} \theta \lambda_L \big( \mathbf{N}_\vartheta \nabla^T \big) \big( \nabla \mathbf{N}_\vartheta^T \big) dV \mathbf{T} + \oint\limits_{\Omega c} q_Q^{n_\Omega} \mathbf{N}_\vartheta dA = 0 \tag{60}$$

Let us introduce the following symbol [16]:

$$\nabla \mathbf{N}_\vartheta^T = \mathbf{B}_\vartheta \tag{61}$$

Thus, the discretized thermal equation will be [16]:

$$\int\limits_{V_c} \theta \rho_L \frac{\partial c_v}{\partial t} \mathbf{N}_\vartheta \mathbf{N}_\vartheta^T dV \mathbf{T} + \int\limits_{V_c} \theta \rho_L c_v \mathbf{N}_\vartheta \mathbf{N}_\vartheta^T dV \frac{\partial \mathbf{T}}{\partial t} + \int\limits_{V_c} \theta \rho_L c_v \mathbf{N}_\vartheta \big( \mathbf{u}^T \mathbf{B}_\vartheta \big) dV \mathbf{T} +$$
$$\int\limits_{V_c} \theta \rho_L \mathbf{N}_\vartheta \big( \mathbf{u}^T (\nabla c_v) \mathbf{N}_\vartheta^T \big) dV \mathbf{T} - \int\limits_{V_c} p\Big( \frac{\partial \rho_L}{\partial t} + \rho_c \frac{\partial}{\partial t}\theta + \mathbf{u}^T (\nabla \rho_L + \rho_c (\nabla \theta)) \Big) \mathbf{N}_\vartheta dV -$$
$$- \int\limits_{V_c} \theta \Xi_L \mathbf{N}_\vartheta dV + \int\limits_{V_c} \theta \lambda_L \mathbf{B}_\vartheta^T \mathbf{B}_\vartheta dV \mathbf{T} + \oint\limits_{\Omega c} q_Q^{n_\Omega} \mathbf{N}_\vartheta dA = 0 \tag{62}$$

taking the form below with constant density and thermal parameters:

$$
\begin{aligned}
&\rho_L c_v \int\limits_{V_c} \theta \mathbf{N}_\vartheta \mathbf{N}_\vartheta^T dV \frac{\partial \mathbf{T}}{\partial t} + \rho_L c_v \int\limits_{V_c} \theta \mathbf{N}_\vartheta \left( \mathbf{u}^T \mathbf{B}_\vartheta \right) dV \mathbf{T} + \\
&-\rho_c \int\limits_{V_c} p \left( \frac{\partial}{\partial t} \theta + \mathbf{u}^T (\nabla \theta) \right) \mathbf{N}_\vartheta dV - \int\limits_{V_c} \theta \Xi_L \mathbf{N}_\vartheta dV + \\
&+\lambda_L \int\limits_{V_c} \theta \mathbf{B}_\vartheta^T \mathbf{B}_\vartheta dV \mathbf{T} + \oint\limits_{\Omega c} q_Q^{n\Omega} \mathbf{N}_\vartheta dA = 0
\end{aligned}
\tag{63}
$$

*3.5. Coupling of the Temperature Fields Developing in the Contacting Bodies and the Lubricant*

Not only the lubricant but also the contacting bodies have to be investigated for the solution of the thermodynamical problem if the boundary conditions according to (30) and (33) are not predefined but have to be determined in the course of solving the thermal problem for coupled (lubricant–contacting body) fields. Two possible routes are available for such coupling. The traditional technique applicable to sub-structures is employed in one of the solutions, retaining the equations set up for both the lubricant and the contacting body at the boundary. This route can be followed easily if the variation solution based on the weak integral of the energy equation is used also for the contacting bodies. According to the other possible route the equation applying to the contacting bodies at the edge is considered to be valid and the variation of the differential equation applying to the lubricant is set up only for the region inside the lubricant. This later possibility is advantageous mainly if an analytical solution is desired to be employed with regard to the temperature of the contacting bodies.

In the finite element method, the unknown fields are attempted to be defined as a linear combination ($\sum_j c_j \varphi_j$) of expediently chosen approximating functions ($\varphi_j$) and unknown constants ($c_j$) in the course of solutions based on variation theory. Constants $c_j$ are defined so as to satisfy the weak form of the original equation. Following the finite element practice, symbol $\mathbf{N}$ will be used for the column vector of the spatial shape functions. While the $\mathbf{T}$ is the column vector of the time dependent $c_j$ "constant" values.

The approximation of the temperature distribution inside the lubricant is [16]:

$$
\vartheta = \mathbf{N}_\vartheta^T \mathbf{T}
\tag{64}
$$

In this case Legendre type elements were used of which shape functions can be grouped as shown below:

- The nodal functions taking the value of 1 at one node of the lattice dividing the region and *0* at the others and varying linearly over the region:

$$
\acute{\mathbf{N}}_i
$$

- The edge functions reaching their maximum at a given edge and having the value of *0* at the others:

$$
\tilde{\mathbf{N}}_i^c
$$

- The surface or side functions having no values at the edges and nodes:

$$
\hat{\mathbf{N}}_i^s
$$

- Plus the internal volumetric bubble function taking any value only inside the element:

$$
\check{\mathbf{N}}_i
$$

- Compiled from these the vector of the approximation functions is obtained in the form below for a 3D element:

$$\mathbf{N}^{\mathrm{T}} = \left[ \acute{\mathbf{N}}^{\mathrm{T}}, \tilde{\mathbf{N}}^{\mathrm{T}}, \hat{\mathbf{N}}^{\mathrm{T}}, \check{\mathbf{N}}^{\mathrm{T}} \right] \tag{65}$$

Let us note that on $S_i$ surfaces only the surface node-, edge-, and side shape functions of the elements of $\mathbf{N}_\vartheta$ have got nonzero values. The shape functions have got three groups: the group of the shape function operating inside the lubricant and groups of those related to surfaces $S_1$ and $S_2$.

$$\mathbf{N}_\vartheta^T = \left[ \mathbf{N}_{\vartheta S_1}^T, \mathbf{N}_{\vartheta V}^T, \mathbf{N}_{\vartheta S_2}^T \right] = \left[ \mathbf{N}_{\vartheta S_1}^T, \mathbf{N}_{\vartheta V, S_2}^T \right] = \left[ \mathbf{N}_{\vartheta V, S_1}^T, \mathbf{N}_{\vartheta S_2}^T \right] \tag{66}$$

$\mathbf{T}$ can also be grouped similarly:

$$\mathbf{T}_\vartheta^T = \left[ \mathbf{T}_{S_1}^T, \mathbf{T}_V^T, \mathbf{T}_{S_2}^T \right] = \left[ \mathbf{T}_{S_1}^T, \mathbf{T}_{V, S_2}^T \right] = \left[ \mathbf{T}_{V, S_1}^T, \mathbf{T}_{S_2}^T \right] \tag{67}$$

Let us introduce the following symbols:

$$\mathbf{N}_{\vartheta V, S_j}^T = \left[ \mathbf{N}_{\vartheta V}^T, \mathbf{N}_{\vartheta S_j}^T \right]; \mathbf{T}_{V, S_j}^T = \left[ \mathbf{T}_V^T, \mathbf{T}_{S_j}^T \right] \tag{68}$$

While the internal shape functions assume *0* value at the contact surfaces, their derivatives will not be equal to *0* at the same locations. On this basis (50) may be written in the following the *i*th surface at $z = z_{Si}$:

$$q_{si} = \theta \lambda_{Lsi} \frac{\partial \mathbf{N}_{\vartheta S_i}^T}{\partial z} \mathbf{T}_{S_i} + \theta \lambda_{Lsi} \frac{\partial \mathbf{N}_{\vartheta V, S_j}^T}{\partial z} \mathbf{T}_{V, S_j} \quad \begin{matrix} i = 1, j = 2 \\ i = 2, j = 1 \end{matrix} \tag{69}$$

Consequently, the surface temperature per (51) in consideration of (69) is:

$$q_{si} = \theta \lambda_{Lsi} \frac{\partial \mathbf{N}_{\vartheta S_i}^T}{\partial z} \mathbf{T}_{S_i} + \theta \lambda_{Lsi} \frac{\partial \mathbf{N}_{\vartheta V, S_j}^T}{\partial z} \mathbf{T}_{V, S_j} \quad \begin{matrix} i = 1, j = 2 \\ i = 2, j = 1 \end{matrix} \tag{70}$$

Let us follow the same approximation method based on least squares:

$$\min \int_{A_c} \frac{1}{2} \left( \mathbf{N}_{\vartheta S_i}^T \mathbf{T}_{S_i} - \vartheta_{si} \right)^2 dA \tag{71}$$

Let us find the minimum of (71) as:

$$\int_{A_c} \frac{\partial}{\partial \mathbf{T}_{S_i}} \left( \frac{1}{2} \left( \mathbf{N}_{\vartheta S_i}^T \mathbf{T}_{S_i} - \vartheta_{si} \right)^2 \right) dA = \int_{A_c} \left( \mathbf{N}_{\vartheta S_i} - \frac{\partial (\vartheta_{si})}{\partial \mathbf{T}_{S_i}} \right) \left( \mathbf{N}_{h S_i}^T \mathbf{T}_{S_i} - \vartheta_{si} \right) dA = 0 \tag{72}$$

Thus [16]:

$$\int_{A_c} \left( \mathbf{N}_{\vartheta S_i} - \int_{A(\hat{x}, \hat{y})} \frac{\partial \mathbf{N}_{\vartheta S_i}}{\partial z} \theta \Lambda_{S_j} dA(\hat{x}, \hat{y}) \right)$$

$$\left( \mathbf{N}_{\vartheta S_i}^T \mathbf{T}_{S_i} - \int_{A(\hat{x}, \hat{y})} \frac{\partial \mathbf{N}_{\vartheta S_i}^T}{\partial z} \theta \Lambda_{S_j} dA(\hat{x}, \hat{y}) \mathbf{T}_{S_i} - \int_{A(\hat{x}, \hat{y})} \frac{\partial \mathbf{N}_{\vartheta V, S_j}^T}{\partial z} \theta \Lambda_{S_j} dA(\hat{x}, \hat{y}) \mathbf{T}_{V, S_j} - \vartheta_{0_s} \right) dA = 0 \tag{73}$$

The arrangement of this results in:

$$
\int\limits_{A_c}\left(\mathbf{N}_{\vartheta S_i} - \int\limits_{A(\hat{x},\hat{y})} \frac{\partial \mathbf{N}_{\vartheta S_i}}{\partial z}\theta\Lambda_{S_j}dA(\hat{x},\hat{y})\right)\left(\mathbf{N}^T_{\vartheta S_i} - \int\limits_{A(\hat{x},\hat{y})} \frac{\partial \mathbf{N}^T_{\vartheta S_i}}{\partial z}\theta\Lambda_{S_j}dA(\hat{x},\hat{y})\right)dA\mathbf{T}_{S_i} -
$$

$$
-\int\limits_{A_c}\left(\mathbf{N}_{\vartheta S_i} - \int\limits_{A(\hat{x},\hat{y})} \frac{\partial \mathbf{N}_{\vartheta S_i}}{\partial z}\theta\Lambda_{S_j}dA(\hat{x},\hat{y})\right)\int\limits_{A(\hat{x},\hat{y})} \frac{\partial \mathbf{N}^T_{\vartheta V,S_j}}{\partial z}\theta\Lambda_{S_j}dA(\hat{x},\hat{y})dA\mathbf{T}_{V,S_j} -
$$

$$
-\int\limits_{A_c}\left(\mathbf{N}_{\vartheta S_i} - \int\limits_{A(\hat{x},\hat{y})} \frac{\partial \mathbf{N}_{\vartheta S_i}}{\partial z}\theta\Lambda_{S_j}dA(\hat{x},\hat{y})\right)dA\vartheta_{0_s} = 0
$$

(74)

Let the equations below apply:

$$
\mathbf{K}_{\vartheta Sii} = \int\limits_{A_c}\left(\mathbf{N}_{\vartheta S_i} - \int\limits_{A(\hat{x},\hat{y})} \frac{\partial \mathbf{N}_{\vartheta S_i}}{\partial z}\theta\Lambda_{S_j}dA(\hat{x},\hat{y})\right)\left(\mathbf{N}^T_{\vartheta S_i} - \int\limits_{A(\hat{x},\hat{y})} \frac{\partial \mathbf{N}^T_{\vartheta S_i}}{\partial z}\theta\Lambda_{S_j}dA(\hat{x},\hat{y})\right)dA
$$

(75)

and, furthermore:

$$
\mathbf{K}_{\vartheta Sij} = -\int\limits_{A_c}\left(\mathbf{N}_{\vartheta S_i} - \int\limits_{A(\hat{x},\hat{y})} \frac{\partial \mathbf{N}_{\vartheta S_i}}{\partial z}\theta\Lambda_{S_j}dA(\hat{x},\hat{y})\right)\int\limits_{A(\hat{x},\hat{y})} \frac{\partial \mathbf{N}^T_{\vartheta V,S_j}}{\partial z}\theta\Lambda_{S_j}dA(\hat{x},\hat{y})dA
$$

(76)

while:

$$
\mathbf{f}_{\vartheta Si} = -\int\limits_{A_c}\left(\mathbf{N}_{\vartheta S_i} - \int\limits_{A(\hat{x},\hat{y})} \frac{\partial \mathbf{N}_{\vartheta S_i}}{\partial z}\theta\Lambda_{S_j}dA(\hat{x},\hat{y})\right)dA
$$

(77)

On the basis of these, the equation originating from the thermal boundary conditions applicable to surfaces $S_1$, $S_2$ may be written as:

$$
\mathbf{K}_{\vartheta Sii}\mathbf{T}_{S_i} + \mathbf{K}_{\vartheta Sij}\mathbf{T}_{V,S_j} + \mathbf{f}_{\vartheta Si}\vartheta_{0_s} = 0 \qquad \begin{matrix} i = 1, j = 2 \\ i = 2, j = 1 \end{matrix}.
$$

(78)

That is:

$$
\left[\mathbf{K}_{\vartheta Sii}, \mathbf{K}_{\vartheta Sij}\right]\mathbf{T} + \mathbf{f}_{\vartheta Si}\vartheta_{0_s} = 0 \qquad \begin{matrix} i = 1, j = 2 \\ i = 2, j = 1 \end{matrix}.
$$

(79)

These equations—together with the thermal equation per (62)—have to be solved by iteration or in parallel with the modification that there $w_Q = \mathbf{N}_{\vartheta V}$ since the conditions expressed by (78) are given at the $S_1$, $S_2$ edges. If the above boundary condition per (78) holds only for the contact surface marked $i$ then $w_Q = \mathbf{N}_{\vartheta VSj}$. In case Equations (62) and (78) are solved in a single iteration cycle, the systems of equations to be solved remain symmetric. In this case only parameters $\mathbf{T}_V$ or $\mathbf{T}_{V,Si}$ are calculated from Equation (62)—depending on whether the above boundary condition model holds for both contact surfaces or only for one of them—and parameters $\mathbf{T}_{sj}$ from Equation (78). If, choosing the parallel solution mode, both equations are desired to be solved in a single step, the thermal equation takes the form below.

$$
\int\limits_{V_c}\theta\rho_L\frac{\partial c_v}{\partial t}\mathbf{N}_{\vartheta V}\mathbf{N}^T_{\vartheta}dV\mathbf{T} + \int\limits_{V_c}\theta\rho_L c_v\mathbf{N}_{\vartheta V}\mathbf{N}^T_{\vartheta}dV\frac{\partial \mathbf{T}}{\partial t} + \int\limits_{V_c}\theta\rho_L c_v\mathbf{N}_{\vartheta V}\left(\mathbf{u}^T\mathbf{B}_{\vartheta}\right)dV\mathbf{T} +
$$

$$
\int\limits_{V_c}\theta\rho_L\mathbf{N}_{\vartheta V}\left(\mathbf{u}^T(\nabla c_v)\mathbf{N}^T_{\vartheta}\right)dV\mathbf{T} - \int\limits_{V_c}p\left(\frac{\partial \rho_L}{\partial t} + \rho_c\frac{\partial}{\partial t}\theta + \mathbf{u}^T(\nabla\rho_L + \rho_c(\nabla\theta))\right)\mathbf{N}_{\vartheta V}dV -
$$

$$
-\int\limits_{V_c}\theta\Xi_L\mathbf{N}_{\vartheta V}dV - \int\limits_{V_c}\theta\lambda_L\mathbf{B}^T_{\vartheta V}\mathbf{B}_{\vartheta}dV\mathbf{T} + \oint\limits_{\Omega c}q_Q^{n\Omega}\mathbf{N}_{\vartheta V}dA = 0
$$

(80)

Or, if adiabatic boundary condition is employed for surface $S_j$:

$$
\int\limits_{V_c} \theta\rho_L \frac{\partial c_v}{\partial t} \mathbf{N}_{\vartheta V,S_j} \mathbf{N}_\vartheta^T dV\mathbf{T} + \int\limits_{V_c} \theta\rho_L c_v \mathbf{N}_{\vartheta V,S_j} \mathbf{N}_\vartheta^T dV \frac{\partial \mathbf{T}}{\partial t} + \int\limits_{V_c} \theta\rho_L c_v \mathbf{N}_{\vartheta V,S_j} \left(\mathbf{u}^T \mathbf{B}_\vartheta\right) dV\mathbf{T} +
$$
$$
\int\limits_{V_c} \theta\rho_L \mathbf{N}_{\vartheta V,S_j} \left(\mathbf{u}^T (\nabla c_v) \mathbf{N}_\vartheta^T\right) dV\mathbf{T} - \int\limits_{V_c} p\left(\frac{\partial \rho_L}{\partial t} + \rho_c \frac{\partial}{\partial t}\theta + \mathbf{u}^T (\nabla\rho_L + \rho_c(\nabla\theta))\right)\mathbf{N}_{\vartheta V,S_j} dV -
$$
$$
- \int\limits_{V_c} \theta\Xi_L \mathbf{N}_{\vartheta V,S_j} dV - \int\limits_{V_c} \theta\lambda_L \mathbf{B}_{\vartheta V,S_j}^T \mathbf{B}_\vartheta dV\mathbf{T} + \oint\limits_{\Omega c} q_Q^{n_\Omega} \mathbf{N}_{\vartheta V,S_j} dA = 0
\tag{81}
$$

Studying the equations obtained if can be found that they are not symmetric thus their solution requires more resources but only one step.

At variation with surface deformations, here the temperature distribution applicable to the surface cannot be incorporated into the thermodynamic equation of the fluid through the use of a cause-and-effect matrix but appears as a direct boundary condition and is determined either by iteration or as a coupled problem in a single step.

Together with the discretized Reynolds equation [15] and with the discretized form of gap size constitute [15] and with the relationship applicable to the parameters of displacement caused by pressure [15] the discretized thermal equation a closely coupled system but the thermal and contact problems must be manageable also separately at the same time.

## 4. Numerical Solution of the System of Equations

Example for solved TEHD problem by the p-version finite element method can be found in the article of Wolff, R., Nonaka, T., Kubo, A. and Matsuo, K., [22] in 1992 since Wolff et al. used the elasto-hydrodynamic problem published by Houpert, L. G. and Hamrock, B. J., [23].

Parameters of the problem:

- Reduced contact radius: $r_{red}$ = 0.0175 mm
- Dimensionless velocity: $U = 2 \cdot 10^{-11}$
- Herzt pressure: $p_h$ = 400 MPa
- Parameters of contacting bodies:
- Modulus of elasticity: $E$ = 200 GPa
- Poisson number: $\mu$ = 0.3
- Density: $\rho$ = 7850 kg/m$^3$
- Coefficient of thermal conductivity: $\lambda_s$ = 52 W/(m·K)
- Specific heat: $c_v$ = 460 J/(kg·K)
- Lubricant properties:
- Lubricant type: Paraffinic oil P-150
- Inlet temperature of lubricant: $T_0$ = 323 K
- Lubricant viscosity at entry: $\eta = 1.539 \cdot 10^{-2}$ Pa·s
- Lubricant density at entry: $\rho$ = 864 kg/m$^3$
- Viscosity-pressure-temperature relationship: modified WLF formula [22]

$$
\lg(\eta_{WLF}) = \lg(\eta_s) - \frac{C_1}{1 + \dfrac{C_2}{[1 - B_1 \cdot \ln(1 + B_2 \cdot p)] \cdot [\vartheta - T_{s_0} - A_1 \cdot \ln(1 + A_2 \cdot p)]}}
\tag{82}
$$

- Coefficient of thermal expansion: $\varepsilon = 6.5 \cdot 10^{-4}$ K$^{-1}$
- Coefficient of thermal conductivity: $\lambda$ = 0.12 W/(m·K)
- Specific heat: $c_v$ = 2000 J/(kg·K)

As it can be seen above, modified WLF formula is used to describe the pressure-temperature relationship of the viscosity. For this task, the specific values of coefficients of modified WLF formula in Equation (82) can be seen in Table 1.

**Table 1.** Coefficients of the WLF formula describing the pressure and temperature dependence of lubricant viscosity [16].

| WLF Coefficient | $T_{S0}$ °C | $A_1$ °C | $A_2$ GPa$^{-1}$ | $B_1$ | $B_2$ GPa$^{-1}$ | $C_0$ | $C_1$ | $C_2$ °C |
|---|---|---|---|---|---|---|---|---|
| | −71.41 | 122.32 | 1.025 | 0.206 | 22.00 | 7.0 | 11.04 | 30.94 |

There were 15 elements in the gap geometry divided along length of contact area. While only a single element was assumed along the thickness for the calculation of temperature.

The elasto-hydrodynamic part of the problem was solved using of optimized damped Newton–Raphson method [16]. The thermodynamical part of the problem was solved by damped Newton–Raphson [16]. The calculated pressure distribution is shown in Figures 3–6. as well as the temperature distribution and the gap size.

Subsequent to the problem of pure rolling contact, the calculations were carried out with 1, 1.9, 2 sliding ratios. Heat generation is much higher in this case, thus the mode of modelling the temperature distribution along the gap strongly influences the results as demonstrated well in the results published by Wolff and his co-authors. In our solution the temperature distribution along the gap was taken fully into consideration by employing fourth degree approximation.

The change in pressure distribution in comparison to the case of pure rolling contact is clearly apparent although a lower effect can be seen in comparison to the pressure distribution calculated by Wolff et al. with the temperature varying along the gap.

The pressure distribution indicated develops at the gap shape and temperature distribution shown in Figures 3–6 demonstrating well the definite temperature rise near the pressure peak shifting towards the upper surface moving more slowly.

The sliding ratio is:

$$S = 2\frac{u_{x2} - u_{x1}}{u_{x2} + u_{x1}} \tag{83}$$

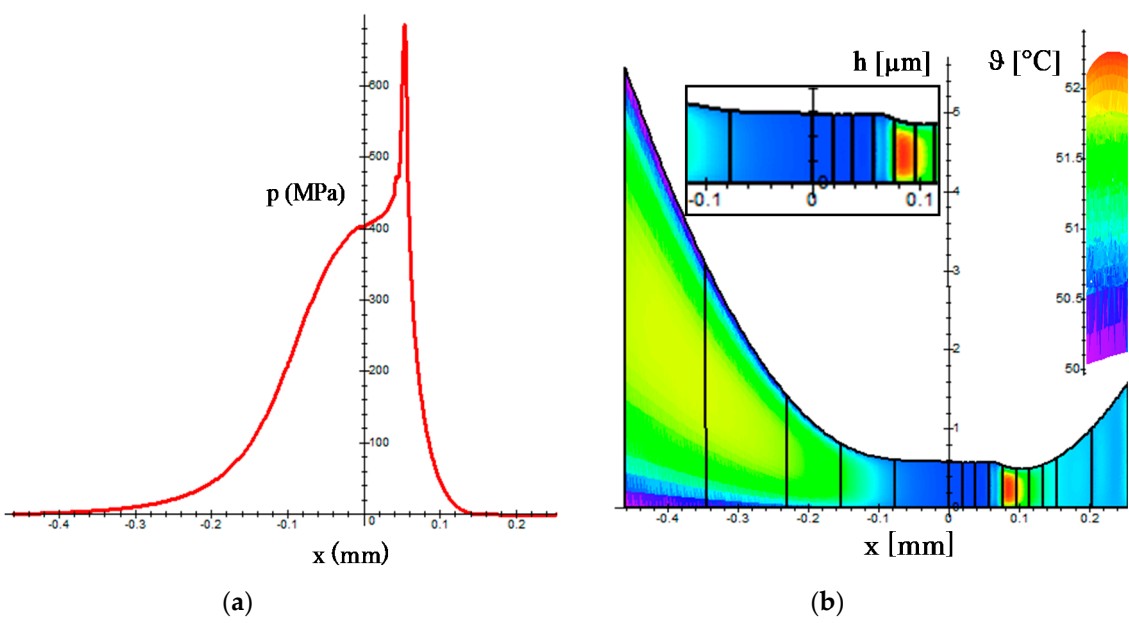

(a)                                                      (b)

**Figure 3.** (**a**) Pressure distribution in the case of pure rolling contact $S = 0$. (**b**) Temperature distribution in the case of pure rolling contact $S = 0$.

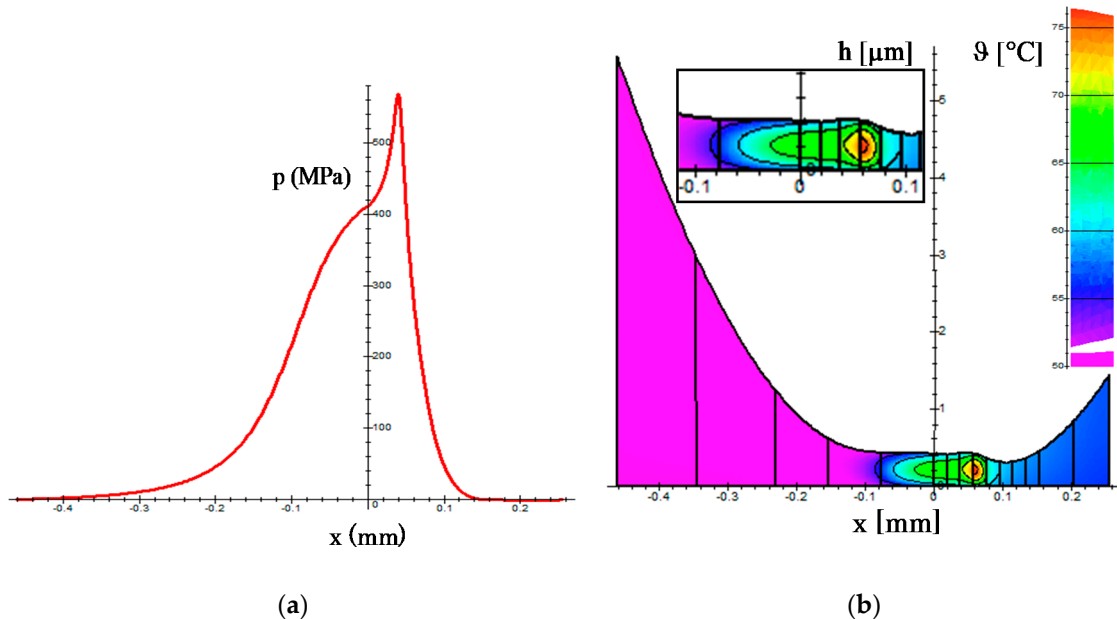

(a)

(b)

**Figure 4.** (**a**) Pressure distribution in the case of rolling-sliding contact S = 1. (**b**) Temperature distribution in the case of rolling-sliding contact S = 1.

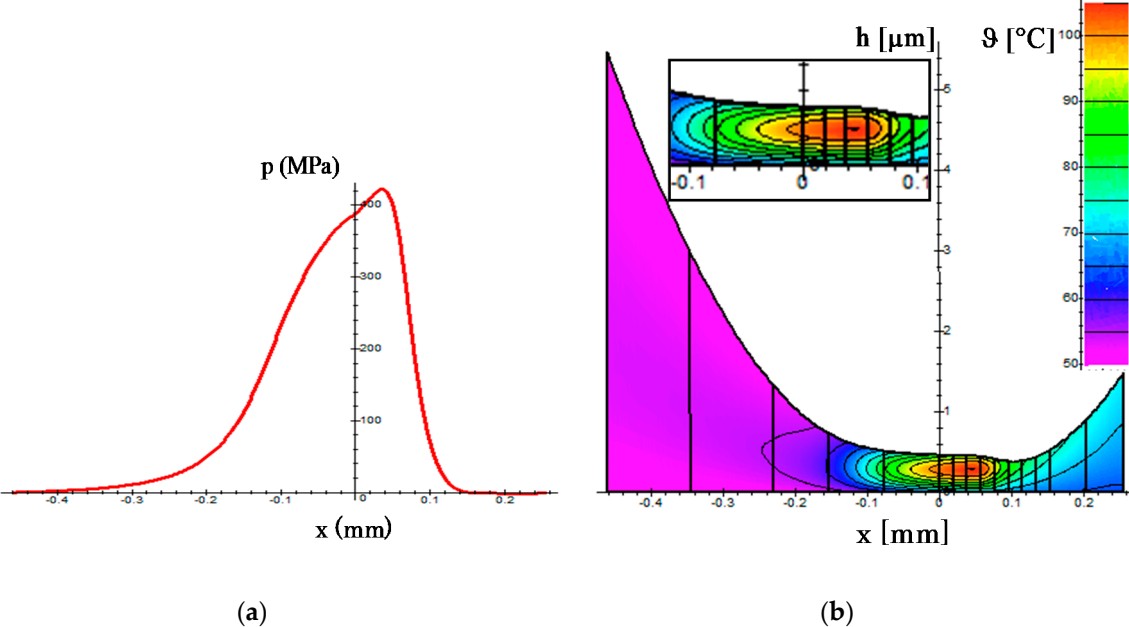

(a)

(b)

**Figure 5.** (**a**) Pressure distribution in the case of rolling-sliding contact *S* = 1.9. (**b**) Temperature distribution in the case of rolling-sliding contact *S* = 1.9.

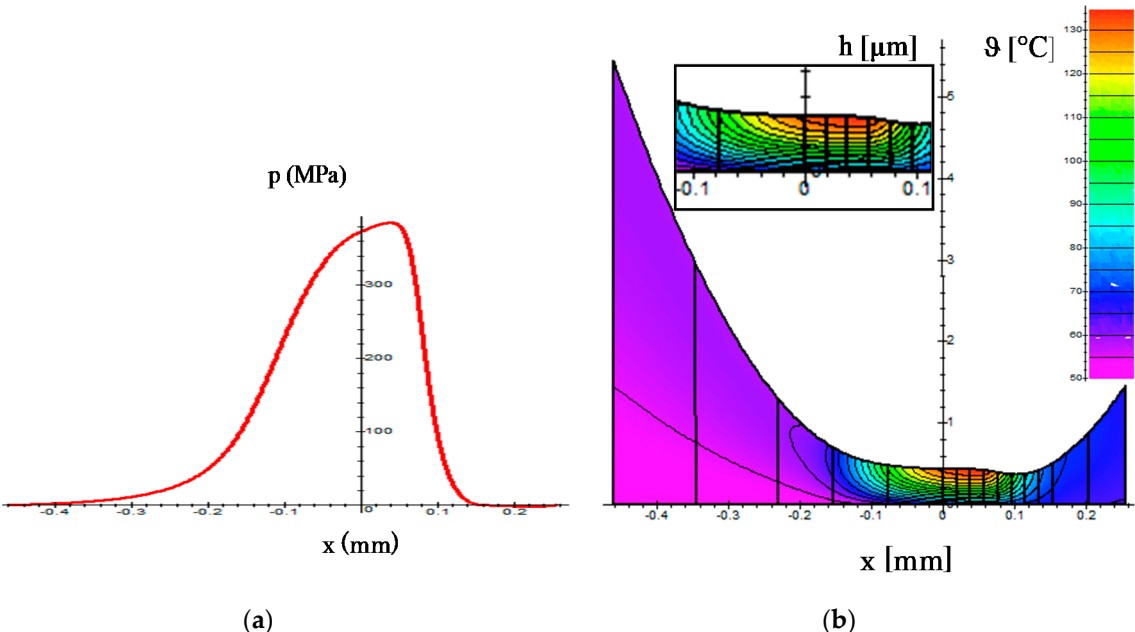

**Figure 6.** (**a**) Pressure distribution in the case of pure-sliding contact *S* = 2. (**b**) Temperature distribution in the case of pure-sliding contact *S* = 2.

## 5. Discussion

The weak integral form of the Reynolds and energy equation have defined and used as a basis of the finite element method. For approximating the pressure and temperature field, Legendre polynomials have been applied as shape functions. Solving the TEHD problems with a p-FEM based procedure allowed us to replace the smooth mesh with a coarse one. On the p-FEM procedure of thermal part of the TEHD problem necessary improvements have been made. This developed method and its application possibility is introduced in this paper. The developed method works in case of Newtonian and non-Newtonian fluid and for both stationery and time dependent case. The thermodynamic model of the fluid–film contact has got non-constant temperature condition inside the film.

As for additional development, it is shown that boundary condition can be redefined as all inlets and outlets can be adiabatic boundaries, therefore, they can be treated uniformly. For the contact surfaces, a substructure model has been defined using analytical solution of the moving heat source where the input temperature data obtained from p-FEM calculation. Instead of an iterative way between the solid and fluid problem, in this paper we present an efficient solution when the thermal model for lubricant and surfaces have been solved as a coupled analysis in one step. In order to handle the cavitation problem a kind of penalty method has been applied on volume fraction parameter that is useful for the p-version finite element method based solution of thermal problem since the continuity equation is valid not only in the contact but in the cavitation zone as well.

The temperature distribution of the fluid film was modelled using higher order FEM approximation. The temperature modelling procedure was coupled to film thickness and pressure calculation. The developed calculation procedure has got a good efficiency for solving TEHL problems. This developed p-FEM procedure can reduce the model size compared to the conventional h-FEM methods and it is also possible to use this process in case of rough surfaces or dynamic load. The calculation performed shows that our p-FEM numerical procedure is valid in case of TEHL computation and the solution is obtained in a stable way.

**Author Contributions:** Conceptualization, S.S.; methodology, S.S.; software, S.S.; validation, S.K.; formal analysis, S.K.; investigation, S.K.; writing—original draft preparation, S.S.; writing—review and editing, S.K.; All authors have read and agreed to the published version of the manuscript.

**Funding:** This research was funded by the Bolyai Fellowship of the Hungarian Academy of Sciences and the by the ÚNKP-18-4 New National Excellence Program of the Ministry of Human capacities.

**Conflicts of Interest:** The authors declare no conflict of interest. The funders had no role in the design of the study; in the collection, analyses, or interpretation of data; in the writing of the manuscript, or in the decision to publish the results.

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
