# Peer review of "Temperature Variation at Solid-Fluid Interface of Thin Film Lubricated Contact Problems"

_processes, doi:10.3390/pr8080922_

Round 1
Reviewer 1 Report
Brief Description of the Work
The objective of the authors is to study the thermo-elasto-hydrodynamic (TEHD) lubrication problems by using the particle finite element method (PFEM) method. More specifically, they used the p-FEM for calculating the film shape, the pressure and temperature distribution, and to model the thermo-hydrodynamic processes supplemented by the (discretized) equations describing the thermal conditions.
Obtained Results
The following two main results have been obtained by the authors:
1) The p-FEM method is a good and stable process to solve the TEHD lubrication problems;
2) The method reduces the model size by decreasing the number of elements (only two-dimensional mesh has to be maintained) and it can also be implemented for treating other problems (such as rough surfaces, dynamically loaded bearings).
General Remarks (R)
R1) Usually, in scientific articles it is customary to specify the acronyms (even if well-known), such as FEM, p-FEM, EHD and TEHD. These terminologies are not specified in the manuscript.
R2) The introduction is written too concisely; it does not provide an exhaustive overview of the results obtained in this research area and currently reported in literature.
R3) The conclusions are too synthetic and not adequately written. Written in this form, the statement of the conclusions are not supported by the results (e.g. the stability of the p-FEM applied to their problem).
Questions (Q)
The theoretical parts, concerning the description of the elastohydrodynamic lubrication and the thermodynamical part of the TEHD, is quite familiar to the reader. The interesting part is Section 4 “Numerical solution of the system of equations” based on the p-FEM.
Q1) This problem has previously been addressed by one of the authors in his work: "Solution method for TEHD fluid contact problem based on p-FEM" that appeared on the website: (https://www.oetg.at/fileadmin/Dokumente/oetg/Proceedings/WTC_2001_files/html/M-16-18-709-SZAVAI.pdf). The authors are kindly invited to specify the added value of the results presented in the current work compared to this latter.
Q2) Generally, in works using the p-FEM for solving the TEHD lubrication problems, in particular in the region of the contacting bodies, the penalty parameter method may get numerical results that are overestimated with respect to the experimental data. Did the authors encounter this problem?
Q3) The authors state that the p-FEM is a stable numerical process for studying the TEHD lubrication problems. This is well-known and generally confirmed by several authors working in the field. However, the authors did not show any test of numerical stability of this method when applied to their problem. May the authors demonstrate the stability of the method for their case this in a supplementary subsection ?
Q4) As known the "face distortion" strongly modify the seals’ thermal behaviour. In the numerical experiments performed by the authors, do they assist to this distortion ?
Conclusions
In my opinion, the work is interesting and it deserves to be published, but not in the present form. The authors are kindly invited to take into account the questions and remarks raised above.
Author Response
General Remarks (R)
R1)
The required abbreviations have been defined in the text.
R2)
The introduction has been modified according to your request.
R3)
The conclusion has been modified according to your request.
Questions (Q)
Q1)
In that work, the structure of the discretized thermodynamic equations was rudimentary, without taking cavitation into account. The treatment of boundary conditions was not presented in that article. Furthermore, we use a completely different cavitation calculation method, as opposed to that presented in that article. In addition, the thermodynamic equations are combined with the developed cavitation equations in this article. So, the calculation itself is carried out by a fully coupled method, as opposed to that referred article where the calculation could only be done in an iterative way between the solid and the fluid thermodynamic problem. Based on the proposed method the analytical solution for the temperature of the contacted surfaces are integrated into the thermal problem of the lubricant as a direct boundary condition.
Q2)
Actually in this paper the determination of the thermal field of the lubricant and the surfaces was our main goal in a new fully coupled way. However in the necessarily adapted cavitation solution, the penalty is applied to the function of the density which is provided the constancy of mass flow also in the cavitation zone while the pressure remains around zero. Contrary to the traditional penalty method the penalty parameter appears only in the volume fraction and the condition of the equation not affected directly by adding a huge value. Since the penalty parameter has no effect in the contact zone, and has role only in the cavitation zone to reduce the volume fraction coefficient to keep constant the mass flow, no any overestimation of the temperature field is expected. Furthermore with this method, we can handle the contact boundary inside a mesh-element in contrast to traditional penalty parameter procedures. So, we did not experience an overestimation to other studies’ results.
Q3)
This is a quite interesting question but for the pressure calculation, not to the temperature field. The governing equation to determine the pressure filed is highly nonlinear in EHD regime but the solution of the temperature filed is quite stable and no any stability issue has to be handled. Anyway the use of Legendre polynomials and the low degree of polynomials used in the approximation of the thermodynamic part significantly reduce the amount of fluctuations experienced during the calculation, as can be seen in the results shown in the figures of the manuscript. Furthermore, as described in Chapter 4, an attenuated Newton-Raphson method was used for ensuring of the stability of the calculation. In the case of EHD, further stabilization was required but it is not presented in this paper. For stabilization the pressure calculation, a procedure was developed in order to find the optimal damping rate for damped Newton-Raphson, which ensured the stability of the calculation. Details of the simple and optimized damped Newton-Raphson procedures used to calculate the total EHD problem will able be found in the references in the appropriate rows of Chapter 4 of the corrected version of the manuscript. “The elasto-hydrodynamic part of the problem was solved using of optimized damped Newton-Raphson method [4]. The thermodynamical part of the problem was solved by simple damped Newton-Raphson[4].”
Q4)
This paper is not about the accurate calculation of the distribution of face distortion, but rather about the treatment of the temperature field. Here we wanted to solve the system of equations for solid and liquid, where solid is assumed to be semi-infinite from a mechanical point of view. In any case, the temperature effects in the face distortion can be incorporated into the δ extent in equation (1).
Reviewer 2 Report
The fundamental change I would like to see in this paper is to focus on the methods used and the model used in the main part of the paper and move much of the mathematics into an appendix, I found the paper very difficult to read because of all of the equations.
I would also like to see the axes on Figures 5-8 expanded around 0 so that the details in that region can be better examined. This is the area where the data is most interesting.
I think this paper makes a contribution to the field, however many will not be interested in all of te details, but more interested in the results from this model.
Author Response
Q1) The fundamental change I would like to see in this paper is to focus on the methods used and the model used in the main part of the paper and move much of the mathematics into an appendix, I found the paper very difficult to read because of all of the equations.
A1) We think that the essence of the article lies in the derivations, because in this way we can prove the correctness of this general procedure, which is also the purpose of the article So we kindly ask to let us keep the equation in the main part.
Q2) I would also like to see the axes on Figures 5-8 expanded around 0 so that the details in that region can be better examined. This is the area where the data is most interesting.
A2) Magnified regions of disputed figures will be included in the corrected version of the manuscript.
Reviewer 3 Report
Topic: Temperature variation at solid-fluid interface of thin
film lubricated contact problems
Interesting and relevant work to record and describe the temperature fluctuation at the solid-liquid interface of thin film-lubricated contact problems.
In this article, an interesting study regarding optimization of the topography of contact surfaces of lubricated systems was presented. Using the developed FE model / procedure, the physical effects like film shape; thickness; pressure and temperature were determined and characterized using the polynomial approximation "p-version". The determination of the surface load conditions is essential in order to be able to calculate the stress field under the surface. A simulation model - method based on the finite element method was developed and described mathematically and numerically, which enables the modeling of thermohydrodynamic processes and the inclusion of discretized equations. It was also possible to describe the thermal conditions. The conducted validation shows the correctness of the presented coupled TEHD method.
The scope of the work is well structured and the method is most often well explained. This work is certainly valuable for the detection and characterization of thermal-flowing processes, which are based on the area of the solid-liquid interface of thin film-lubricated contact problems.
Before acceptance, it is recommended to clarify the following questions or ambiguities: Figures 3 and 4 are missing or is the numbering incorrect? Please correct!
In this article, I miss the implementation and testing of the method / model based on the solution of a specific application to evaluate the potential and possible applications of the technology/method and its system of equations as well as to improve or modify it.
Author Response
Q1) Before acceptance, it is recommended to clarify the following questions or ambiguities: Figures 3 and 4 are missing or is the numbering incorrect? Please correct!
A1) The numbering has been corrected in the manuscript. Thank you!
R1) In this article, I miss the implementation and testing of the method / model based on the solution of a specific application to evaluate the potential and possible applications of the technology/method and its system of equations as well as to improve or modify it.
A2) In this article, we focused on the general definition of a direct algorithm solution procedure for the TEHD problem. Based on theoretical considerations for the most part, we have written a generally applicable stable system, which provides an opportunity to study arbitrary applications in the future. Based on your suggestion we will focus on the implementation in our future work.
Round 2
Reviewer 1 Report
The authors took into account the suggestions made in my previous report and answered to my questions in an exhaustive way.
Reviewer 2 Report
While I still think the derivations should be included in an apendix, I will yield to the authors although I think this will decrease the overall readership of the article.